# Aromatypicity of Austrian Pinot Blanc Wines

**DOI:** 10.3390/molecules25235705

**Published:** 2020-12-03

**Authors:** Christian Philipp, Phillip Eder, Sezer Sari, Nizakat Hussain, Elsa Patzl-Fischerleitner, Reinhard Eder

**Affiliations:** Höhere Bundeslehranstalt und Bundesamt für Wein- und Obstbau, Wienerstraße 74, 3400 Klosterneuburg, Austria; phillip.eder@weinobst.at (P.E.); sezer.sari@weinobst.at (S.S.); h.nizakat@gmail.com (N.H.); elsa.patzl-fischerleitner@weinobst.at (E.P.-F.); Reinhard.Eder@weinobst.at (R.E.)

**Keywords:** wine flavour, esters, free monoterpenes, origin, vintage, descriptive analysis

## Abstract

Pinot blanc is a grape variety found in all wine-growing regions of Austria. However, there are only few scientific studies which deal with the aroma of wines of this variety. In the course of this project, the relationship between aroma profile and the typicity of Austrian Pinot blanc wines was studied. The aim was to describe the typicity and to find significant differences in aroma profiles and aroma descriptors of typical and atypical Pinot blanc wines. Since the typicity of a jointly anchored prototype is embedded in the memory, typical attributes for Austrian Pinot blanc wines were first identified by consumers and experts or producers. According to this, 131 flawless commercial Austrian wines of the variety Pinot blanc of the vintages 2015 to 2017 were analysed for more than 100 volatile substances. The wines of the vintages 2015 to 2017 were judged by a panel of producers and experts for their typicity; furthermore, the wines of the vintage 2017 were also evaluated by a consumer panel and a trained descriptive panel. Subsequently, typical and atypical wines were described by the trained descriptive panel. It was found that Pinot blanc wines typical of Austria showed significantly higher concentrations of the ester compounds ethyl hexanoate, ethyl butanoate, ethyl octanoate, ethyl decanoate, methyl hexanoate, hexyl acetate and isoamyl acetate, while atypical wines had higher concentrations of free monoterpenes such as linalool, trans-linalool oxide, nerol oxide, nerol and alpha-terpineol. The sensory description of typical Pinot blanc wines was significantly more pronounced for the attribute “yellow pome fruit”, and tended to be more pronounced for the attributes “green pome fruit”, “pear”, “walnut”, “pineapple”, “banana” and “vanilla”, while the atypical Pinot blanc wines were described more by the attribute “citrus”. These findings could help to ensure that, through targeted measures, Austrian Pinot blanc wines become even more typical and distinguish themselves from other origins such as Germany or South Tyrol through a clear concept of typicity.

## 1. Introduction

The popularity of Austrian wine has gradually increased at home and abroad over the past decades. This success is primarily due to wines of the varieties Gruener Veltliner, Riesling, Sauvignon blanc and Blaufraenkisch from different wine growing regions [1,2]. In fact, the Austrian wine landscape is much more heterogeneous. For example, 26 white and 14 red varieties are approved as Austrian quality grape varieties [3].

“Weißburgunder” (Pinot blanc) is the sixth most important white wine variety in terms of the area planted with vines and is represented in all wine-growing regions [4]. In international comparison, Pinot blanc is mainly grown in Germany, Austria, Italy and France. In 2015, there was a total of 14,834 ha planted with of Pinot blanc worldwide. This corresponds to a share of less than 0.2% of a total global vine area of about 7.5 million hectares. Nevertheless, the cultivation of this vine variety is economically relevant. However, not much is known about the aroma of wines of this variety. Pinot blanc were classified as a neutral grape variety and distinguished from aromatic grape varieties (for example, Muscat, Traminer, etc.) and semi-aromatic varieties (for example, Rheinriesling and others) [5]. The importance of monoterpenes for the typicity of German Pinot blanc was emphasised [6], and the pear aroma in Pinot blanc wines was characterised [7]. The influence of selected oenological decisions on the aroma of wines of this variety was studied in several cases [8,9,10].

This publication deals with the aroma of Austrian wines of the variety Pinot blanc of different vintages and origins and its relation to the typicity of the wines. A wine is typical if some of its own characteristics are recognisable, which belong to a defined type and are distinguishable from other types [11]. The concept of typicity thus thrives on the existence of a common prototype, embedded in memory, which reflects the image of all previous experiences with wines of the same type [12]. A distinction is made between wine variety typicity, the typicity of PDOs (products with a protected designation of origin) and the typicity of wine styles (special oenological measures, such as wood barrel storage, biological acid reduction, etc.) [11]. 

A number of articles has dealt with the question of how to measure the level of typicity. For example, it was suggested that typicity should be considered both from the point of view of the producers of the product and from the point of view of consumers who are representative of a product [13,14], or a global approach was pursued concerning the assessment of typicity [15]. Therefor it was recommended not to use the word “typicity” because it has no clear definition. Examiners were asked to imagine explaining a good example of a Chardonnay wine to a friend (for example) [15]. For each sample, they had to answer the following question: “Do you think this wine is a good or bad example of a Chardonnay wine?” The scale was unstructured and anchored with a “very bad example” at the left end and a “very good example” at the right end. This method has already been widely used to assess the typicity of wines made from different grape varieties, for example, Chardonnay [16], Sauvignon blanc [17], Riesling [18] and Melon de Bourgogne [16,19]. Furthermore, a method for assessing typicity was described in several steps [20]. The first step was to ensure the existence of a common understanding within the panel of what constitutes a typical wine in the category to be evaluated (for example, for Austria DAC (Districtus Austriae Controllatus)). A second step was the mandatory linking of this common concept of typicity with membership in the typicity category (for example, membership of DAC producers). The third step was to evaluate the relationship between the typicity of the wines in the global environment, that is, the differentiation from other products. The fourth and final step was therefore the complete characterisation of wines that are considered typical. Furthermore, in the common literature, a method for evaluating the typicity of PDO wines with experts from the wine industry was also described, and it was shown that experts from the terroir of the wines to be evaluated are best suited for this purpose, while the evaluation by experts outside the region led to different results [21].

In the course of the present work, 131 Austrian wines of the Pinot blanc variety were analysed. The aim was to characterise the typicity of Austrian Pinot blanc wines with regard to their aroma profile. For this purpose, the typicity according to [16] was evaluated, the recommendation of [13,14] was taken into account, and the method for evaluating the typicity in several steps according to [20] was modelled. As Pinot blanc is not the leading variety in any wine-growing region in Austria, no regional types were developed, but attention was paid to the differences between the field of the vision of producers and consumers. Wines of different vintages and origins were evaluated for their typicity and then classified into typical and atypical wines, and differences in the aroma profile of the wines were analysed. 

## 2. Results

### 2.1. Vintage and Origin Differences in the Aromatic Concentrations of the Wines Examined

Table 1 shows the concentrations of the volatile substance groups of the different origins and vintages, while the average concentrations of all quantifiable individual compounds and the significant effects of vintage, origin and the interaction between vintage and origin are shown in Appendix A.

In total, a significant difference between vintages was found for 40 of the 78 quantifiable aromatic compounds in Pinot blanc, but only 16 compounds showed a significant difference with regard to the different origin (generic wine regions). Furthermore, a significant interaction (origin x vintage) was identified for seven compounds. With regard to the aroma groups, the sum of the C6-components, the sum of the ethyl esters and the sum of the methyl esters were influenced by the vintage year and, in the case of the free monoterpenes and aromatic esters, by the origin (Table 1).

Regarding to free monoterpenes, the highest quantities were found for linalool and geraniol. For example, the range for free linalool concentrations in white wines was between <5 and >300 µg/L [22,23]. Therefore, the detected concentrations in Pinot blanc were found in the lower range of the indicated bandwidth. The same applied to other detectable monoterpenes [24]. It was not surprising that significant differences in free monoterpenes concentrations were found between the averages of the individual vintages, since the genesis of monoterpenes in wines depends on many factors, including the grape variety and the cultivation and weather conditions [22]. The fact that these differences do not appear more clearly and consistently is also due to the complexity of the interrelationships of the factors just mentioned as well as influencing factors of the release from sugar-bound precursors [25] and heterogeneity of the sample material regarding different wine production. Furthermore, it should be noted that, although no significant regional effects with respect to individual compounds were found, with respect to the sum of free monoterpenes, significant effects were detected. Wines of Styrian origin showed higher values than wines of other origins. This fact is interesting because the Huglin indices of Styrian origin are lowest compared to other origins (Appendix A). 

The C6 components (*Z*)-3-hexen-1ol and hexanol were also quantified in the Pinot blanc wines. Significantly higher contents of (*Z*)-3-hexen-1ol were found in the wines from Styria compared to wines from Burgenland and Vienna. For both substances, as well as for the sum of both substances, significant differences were found with regard to the vintage groups. C6 alcohols were used as variety markers for assessing the origin of wine, in particular Spanish grape varieties and origins [26]. The ratios of (*Z*)-3-hexen-1ol/(*E*)-3-hexen-1ol as well as 1 hexanol/(*Z*)-3-hexen-1ol and 1 hexanol/(*E*)-3-hexen-1ol were identified as markers of origin. In the course of the present study, the (*E*)-3-hexen-1ol content was not quantified; however, significant differences were found in the 1-hexanol/(*Z*)-3-hexen-1ol ratio in relation to the generic vineyards. The average ratio varied from 10.1 (Styria) to 14.0 (Burgenland). In between, the ratio for Lower Austria was 12.8 and for Vienna 13.4, and the ratio found was significantly lower in Styrian wines than in wines from the rest of Austria. 

Higher alcohols, volatile fatty acids and ester compounds are aroma compounds that are strongly related to amino acid metabolism [27]. Both regional effects and vintage effects are to be expected, but, above all, oenological decisions, such as the choice of yeast and yeast nutritional salt, can significantly override these effects [28]. While significant regional effects were observed for isobutanol and 1-butanol in the case of the higher alcohols, a significant vintage effect was observed for volatile carboxylic acids in the case of butyric acid, hexanoic acid and octanoic acid. The role of esters in Pinot blanc was described in [29] and was discussed in detail. For 20 ester compounds, significant differences in the concentrations of the wines of the different vintages were found, whereas for 10 ester compounds, significant differences were found in terms of origin. With regard to the sums of the ester groups, the ethyl esters and methyl esters were significantly influenced by the vintage, while the aromatic esters were influenced by the origin (significantly higher for wines from Styria). For ethyl isovalerates, ethyl heptanoates and isobutyl propionates, significant interactions between vintage and origin were found (Appendix A). 

On the part of the C13-norisoprenoids, 1,1,6-trimethyl-1,2-dihydronaphtalene (TDN) and vitispiran were quantified. It should be noted that some of the C13 norisoprenoids are sugar-bound as precursors, and the detection of free concentrations is only of limited use. Various factors, including release during wine storage and regional differences were described [30], which in turn may have various causes related to climate [31], clone selection [32] and vineyard management [18,33,34]. Among the Pinot blanc varieties examined, TDN had a significant regional effect. The concentrations of TDN was significantly lower in wines from Styria than in those from Vienna and Burgenland (Appendix A).

3-isobutyl-2-methoxypyrazine was detectable in the Pinot blanc wines examined, but below the limit of quantification (LOQ = 2 ng/L).

The group of carbonyl compounds is very heterogeneous. While diacetyl is mainly formed by lactic acid bacteria and thus serves as an indicator of biological acid degradation [35,36,37], higher contents of furfural, 3.5-dimethoxy-4-hydroxyacetophen (acetosyringone), 5-methylfurfural, 5-acetoxymethyl-2-furalaldehydes (5-acetoxymethylfurfural) and syringaldehydes are associated with wood barrel storage [38,39]. The above-mentioned compounds are mainly in the context of oenological decisions. Regional, but also vintage differences with regard to the above-mentioned volatile substances may be an indication of processing differences. In fact, all wines of Leithaberg origin (Leithaberg DAC Pinot blanc) are required by law to use a moderate amount of wood content (large wood barrel), which in turn could explain the significantly higher contents in furfural, because 18 of the 24 wines from Burgenland came from Leithaberg [40].

Furthermore, the lactones delta-decalactone, (*E*)-whiskey-lactone and (*Z*)-whiskey-lactone were determined in commercial Pinot blanc wines. While delta-decalacton occurs in white [41,42,43], red [44] and sweet wines [45] even without wooden barrel storage, (*E*)-whiskey-lactone and (*Z*)-whiskey-lactone are mainly associated with wooden barrel storage [46]. 

Volatile phenols are hardly detectable in grape must [47], but a number of compounds can be found in wines. The activity of yeast and bacteria, the wine maturation and especially the storage in wooden barrels lead to a change in the profile of these compounds in the wine. Glycosides of 4-vinylguaiacol, 4-vinylphenol and eugenol could exist in some grape varieties [48]. At best, hydroxycinnamic acids, such as 4-hydroxycinnamic acid, para-coumaric acid or ferulic acid, are the most important precursors for these compounds [49]. Thus, differences in vintage and origin are due both to oenological decisions (choice of yeast, storage in wooden barrels, etc.) and to different amounts of precursors. 

Appendix BFigure A1 also contains a heat map in order to cluster regional and seasonal effects.

### 2.2. Description of Typical Wines of the Pinot Blanc Variety by Different Panels

Consumers and producers were interviewed to generate typical attributes for Austrian Pinot blanc wines. The frequency of citation correlates with the font size of the attributes in the word clouds displayed (Appendix A). The diagrams normalised to the number of respondents show only slight differences in the quality of the terms, but visible differences in the frequency of citations. 

Table 2 shows the percentage response to an attribute of the respective interviewed group. The attributes “walnut”, “almond”, “banana”, “meadow flower” and “oak” were mentioned significantly more often by producers and experts, while the attributes “apricot”, “green apple”, “yellow apple”, “lemon”, “elderflower” and “vanilla” were mentioned significantly more often by consumers.

### 2.3. Typicity Assessment

Figure 1a shows the typicity assessment of the evaluated Pinot blanc wines in terms of regional distribution (generic wine regions), evaluated by a panel of experts consisting of Pinot blanc producers. The determined typicity for wines from Lower Austria was significantly higher than the typicity for wines from Styria. The overall typicity assessed by the expert panel showed no significant differences between the vintage years (Figure 1b). Typicity varied on a 10 cm scale (0 “atypical” and 10 “very typical”) from 2.61 to 8.29 in the 2015 vintage, from 2.82 to 7.18 in the 2016 vintage and from 2.32 to 8.20 in the 2017 vintage. In addition, 2017 wines were evaluated by three different panels (an expert panel (consisting of Pinot blanc producers), consumer panel, and trained descriptive panel). The scattering of results between samples was slightly larger in the trained panel and the expert panel than in the consumer panel. In addition, the trained panel rated the wines significantly higher than the other two panels (Figure 1c). However, the results showed a significant positive correlation (consumer panel to expert and producer panel: R^2^ = 0.595; *p* < 0.01; consumer to trained panel: R^2^ = 0.563; *p* < 0.01; expert and producer panel to trained panel: R^2^ = 0.380; *p* < 0.05). 

### 2.4. Differences in Aroma Profile between Typical and Atypical Pinot Blanc Wines

Appendix A shows the number of typical (typicity > 6), medium typical (4 < typicity < 6) and atypical rated wines (typicity < 4) of each tasting. Table 3 shows compounds whose concentrations were significantly higher in typical (typicity > 6) assessed Pinot blanc wines compared to atypical (typicity < 4) assessed wines. In total, this was found for 21 compounds. It is noticeable that 14 ester compounds were found among them. Ethyl hexanoate was the only compound for which the significant differences of the expert panel review withstood all three years. In the case of ethyl octanoate, the vintages 2016 and 2017 and in the case of isoamyl octanoate and methyl vanillates, the vintages 2015 and 2016 showed significant differences in the expert panel review. In addition, a significant difference was found for the compounds ethyl hexanoate, ethyl butanoate, ethyl octanoate, ethyl decanoate, methyl hexanoate, hexyl acetate and isoamyl acetate in the case of considering wines from all vintages together. 

Conversely, 22 compounds significantly occurred at least once in atypically assessed Pinot blanc wines in higher quantities than in typical Pinot blanc wines (Table 4), including a number of monoterpene compounds, such as linalool, trans-linalooloxide, neroloxide, nerol and alpha-terpineol. Higher concentrations of linalool were found to be rather negative for the typicity assessment by the experts in all three vintages. 

Furthermore, fermentation by-products, such as the higher alcohols propanol, isobutanol and isoamyl alcohol, as well as the carboxylic acids isovaleric acid and hexanoic acid, were put on the list of those compounds for which higher concentrations had rather negative effects on the typicity. A number of volatile phenols, including eugenol, trans-isoeugenol, 2-methoxy-4-propylphenol and 4-methylguaicol, had at least one negative effect on the assessment of typicity, while a number of other volatile phenols, such as guaiacol and acetovanillones, had at least one positive effect on the assessment of typicity. The role of lactones is also remarkable. In contrast, higher concentrations of delta decalactone achieved significantly higher typicity ratings in 2016, and, as interpreted from the consumer’s panel, higher concentrations of (*E*)-whiskey-lactone had multiple negative effects.

### 2.5. Description of Typical and Atypical Wines

Eight typical (typicity > 6) and eight atypical (typicity < 4) wines of the 2017 vintage were described by the trained panel. Appendix A shows the number of typical and atypical wines of each tasting. Since a balanced number of typical and atypical wines should be tasted, the mean values of the consumer and expert or producer panels were used for the selection of the wines. This resulted in eight wines each. The eight typical wines had also been judged as typical by the trained panel. The eight atypical wines were described by the trained panel as atypical (five wines) and medium typical (three wines). However, the description was made by the trained panel. Figure 2 shows the spider diagram of the mean values of typical and atypical wines. Non-parametric tests were used to evaluate the sensory analysis, as the usual three-way ANOVA was not possible because the data were not normally distributed. A significant influence was only found for “yellow pome fruit”. The atypical wines show lower average values than the typical wines for all flavours, except for “citrus”. The citrus aroma was found to be atypical. The ratings of the individual wines are shown in Appendix A.

## 3. Discussion

In the course of the present study, the scientific question of describing the typicity of Austrian Pinot blanc wines was addressed. The concept of typicity thrives on the existence of a common prototype, anchored in memory, which reflects the image of all previous experiences with wines of the same type [12,50]. In the course of the study, the sensory space for Austrian Pinot blanc wines was defined. Austrian Pinot blanc wines have been described with the aroma attributes “pear”, “apple”, “walnut”, “almond”, “citrus” and “pineapple” and, in the special case of ripened Pinot blanc, with “vanilla”. This is in agreement with the statements of [51] and is comparable with the description of Pinot blanc wines from South Tyrol [52] and Germany [53]. The suggestion that typicity should be assessed by consumers and producers as well as by experts is also mentioned in [13,14]. This request has been taken into account in the present work. There were differences in the following attributes: “walnut”, “almond”, “banana”, “meadow flower”, “oak”, “apricot”, “green apple”, “yellow apple”, “lemon”, “elderflower” and “vanilla”. Consumers preferred the fruity attributes, while producers and experts preferred the nutty flavours. On the other hand, it is interesting to note that in the typicity assessment of wines, no significant differences between the judgements of consumers and the ratings of producers were found. This is probably due to the training of the test persons with previously determined word clouds (Appendix A). The trained descriptive panel, which was subsequently used to characterise typical and atypical Pinot blanc wines, rated the typicity of the Austrian wines significantly higher, but the correlation factor among the three panels in general was significantly positive at over 0.38, making the further procedure with the selection of typical and atypical wines and subsequent sensory characterisation by the trained panel plausible. 

Pinot blanc, which is typical for Austria, showed higher concentrations of medium-chain ester compounds, such as ethyl hexanoate, ethyl octanoate and ethyl decanoate. This has already been described in [7,9], and it should be seen as a meaningful implication [54,55]. The enrichment (increase) of a compound does not automatically lead to an intensification of the desired aroma. Furthermore, as already mentioned in [29], the ratio of methyl ester to ethyl ester in Pinot blanc wines is low compared to wines of other varieties. This is probably due to the thin skin of Pinot blanc and the resulting low pectin content. Higher methyl ester compounds (methyl hexanoate, methyl octanoate and methyl decanoate) nevertheless had a positive effect on the typicity assessment of the wines. It was already shown that minor ester compounds contribute to the aroma of wines [56]. At best, this study has shown that ester compounds are important for the typicity of Austrian Pinot blanc wines.

The importance of quantifying free monoterpenes in the sub-threshold range was emphasised by [57]. Rather negative evaluation of free monoterpenes in connection with Pinot blanc typicity is interesting for several reasons. The importance of monoterpenes for the typicity of German Pinot blanc was emphasised by [56]. Higher concentrations of free monoterpenes were associated with the fresh fruity wine type (lemon, green apple, and grapefruit) [58]. It could also be shown that atypical wines of the 2017 vintage had a higher value for the citrus fruit descriptor. Higher concentrations of monoterpenes were consistently found in wines of Styrian origin, which in turn are advertised as fresh and fruity, especially in the area of regional wines (Austrian regulation of DAC Styria). The Styrian origin consistently had the lowest Huglin index. Due to the northern location, this can also be assumed for the wine-growing areas from Germany. The wines of Styria also stood out due to significantly worse typicity ratings. This does not necessarily mean that the wines from Styria are more atypical, but that they are different. Besides higher contents of free monoterpenes, a significantly lower ratio of 1-hexanol/(*Z*)-3-hexen-1ol, significantly lower TDN concentrations and significantly higher contents of aromatic esters were remarkable. 

Future work could focus on confirming the differences between Styrian Pinot blanc and Pinot blanc from Eastern Austria on the one hand, and, on the other hand, on the differences with other countries of origin, such as South Tyrol or Germany. In this respect, however, not only commercial wines should be used, but also standardised micro-vinifications. 

## 4. Materials and Methods 

### 4.1. Samples

Twelve bottles each of 131 commercial wines of the variety Pinot blanc from a total of 64 Austrian wineries were collected for the study. The distribution of the wines is shown in Appendix A. The average Huglin indices were calculated in cooperation with the ZAMG (Central Institute for Meteorology and Geodynamics) for the nearest climate station. According to this, the annual averages for Austria fluctuated from 2090 (2016) to 2180 (2015). 

The Pinot blanc wines were purchased from the winegrowers at the earliest six weeks after bottling, from April to October after the harvest, depending on how the wine was matured, and stored at 4 °C until analysis. All wines were described by the winegrowers as being typical of variety and vintage and were vinified, according to the producer’s statement, without any varietal or vintage blends. All wines were classified as quality wine according to the Austrian Wine law (BGBl. I Nr. 111/2009) [59]. Those wines should therefore be regarded as being free from defects, typical for variety and for vintage and, regarding Austrian quality wine, of a minimum quality. Six bottles each were used for sensory evaluation and six bottles for analytical characterisation.

### 4.2. Analysis of Volatile Substances

#### 4.2.1. Materials

All chemical standards and deuterated standards were obtained from Sigma Aldrich (St. Louis, MO, USA), Merck (Darmstadt, Germany), Carl Roth (Karlsruhe, Germany), Deutero GmbH (Kastellaun, Germany) and EQ-Laboratories (Augsburg, Germany) and showed the maximum available concentration (90%–99.5%). All standards that are not commercially available were produced by self-synthesis [60,61]. The standards for the analysis of C13 norisoprenoids were received from the Technical University of Braunschweig [62].

The sodium chloride (99.8%), used for the aroma analyses, came from Carl Roth (Karlsruhe, Germany) and the sodium hydroxide (10 mol/L) from Sigma Aldrich (St. Louis, MO, USA). Ethanol (99%) was purchased from AustrAlco (Österreichische Alkoholhandels-GmbH, Spillern, Austria), methanol (100%) from VWR Chemicals (Darmstadt, Germany), dichloromethane (99.9%) from Carl Roth (Karlsruhe, Germany) and chloroform (99.8%) from Sigma Aldrich (St Louis, MO, USA).

#### 4.2.2. Analysis Design and Instrumentation

Of the 131 wines, a total of 102 aroma substances of the aroma groups esters, higher alcohols, carboxylic acids, carbonyl compounds, subtle wood aroma substances (lactones, volatile phenols), free monoterpenes and free C13 norisoprenoids were determined using six different GC-SIM-MS methods. Two gas chromatographs from Agilent Technologies (Santa Clara, CA, USA) were used for the analysis of the various volatile substances. The first system, consisting of a 6890 N GC system with a 5975 Inert Mass Selective Detector and a CTC Analytics Autosampler (Zwingen, Switzerland), was equipped with a ZB-Wax plus (length: 60 min, I.D.: 0.25 mm, df = 0.25 µm) from Phenomenex (Torrance, CA, USA) and was used for the analysis of the main aroma substances and C13 norisoprenoids. The second system, consisting of a 7890A GC system with a 5975C Inert MSD with Triple Axis Detector and a CTC Analytics Autosampler (Zwingen, Switzerland), was used for the analysis of major and minor ester compounds, methoxypyrazines and free monoterpenes as well as for the analysis of various flavouring substances. This system was equipped with a ZB-5MS column (length: 60 min, I.D.: 0.25 mm, df = 0.25 µm) from Phenomenex (Torrance, CA, USA).

The wines were analysed between November and December of the year following the harvest. The samples were taken from the cold store and brought to room temperature. All analyses were performed in duplicate from two independent wine bottles. Information on validation and calibration is given in Appendix A.

#### 4.2.3. Quantification of Main Flavouring Substances

The determination of the quantitative main aroma compounds, such as relevant higher alcohols, relevant short and medium-chain carboxylic acids, some carbonyl compounds and ester compounds, was carried out using a partial SIDA-HS-SPME-GC-MS method [58].

#### 4.2.4. Quantification of Ester Compounds

The quantification of 33 minor and major ester compounds was performed by partial SIDA-SPME-GC-SIM-MS [29,63].

#### 4.2.5. Quantification of Free Monoterpene Compounds

The determination of 15 free monoterpene compounds was performed by the HS-SPME-GC-SIM-MS method using a 65 µm, polydimethylsiloxane/divinylbenzene (PDMS/DVB)-fused silica/SS Fiber Core (SUPLECO, Bellefonte, PA, USA) fibre. A total of 1.5 g salt (NaCl), 5 mL sample and 10 µL internal standard (3,4-dimethylanisole in methanol, nominal concentration of 35 µg/L) were added to a 20 mL headspace vial. The sample was preincubated in the autosampler unit for 0.1 min and then extracted at an incubation temperature of 50 °C with a magnetic stirrer at 500 rpm for 30 min. The fibre was then injected “splitless” into the gas chromatograph and the extracted sample desorbed for five min at 250 °C.

The injection temperature was 250 °C, and helium (flow rate: 1.2 mL/min) was used as carrier gas. The initial temperature in the oven was 50 °C for three min. Subsequently, it was heated once at a rate of 1 °C per minute to 92 °C, held for ten min, and then heated at a rate of 5 °C per minute to 127 °C. Afterwards, the temperature was heated up to 260 °C at a rate of 40 °C per minute and held again for 4.6 min. The transfer line for the mass-selective detector had a temperature of 250 °C. The measurements were performed in single-ion mode (EI^+^, 70 eV). The running time was 70 min. The validation was carried out in a Pinot blanc regular wine, the five-step calibration in buffered artificial wine.

#### 4.2.6. Quantification of 2-Isobutyl-3-methoxypyranzine

The quantification of 2-isobutyl-3-methoxypyranzine was performed according to the method described by [64] using the SIDA-SPME-GC-SIM-MS method.

#### 4.2.7. Quantification of Selected C13 Norisoprenoids

The quantification of the C13-norisoprenoids 1,1,6-trimethyl-1,2-dihydronaphtalene and vitispiran isomer mixture was performed according to the SIDA-SPME-GC-SIM-MS method of [61].

#### 4.2.8. Quantification of Other Compounds

Wood flavourings, volatile phenols, lactones and some other volatile substances were determined by SPE-GC-SIM-MS. The adapted method is based on the methods of [65,66]. A total of 10 mL of the samples was mixed in a volumetric flask with 50 µL internal standard (3,4-dimethylanisole, nominal final concentration of 10 µg/L) and pre-text. SPE tubes (LiChrolut EN 200 mg) from Merck (Darmstadt, Germany) were cleaned twice with dichloromethane-chloroform (mixture 2:1) and twice with methanol. The SPE tubes were conditioned twice with 1 mL buffered artificial wine (prepared in accordance with [65]). The samples were then applied under gravity, dried for 15 min under vacuum, and then eluted into new test tubes with 2 times the 0.75 mL dichloromethane-chloroform mixture. The eluate was dried with a spatula tip of sodium sulphate, evaporated under air flow to approx. 0.3 mL, and transferred into 200 µL microvials. A total of 1 µL was injected at 260 °C in split mode (5:1). Helium was used as carrier gas (flow rate 1 mL/min). The temperature program began with a 5-minute holding phase at 50 °C and a subsequent increase of 4 °C/min to 120 °C. This was followed by an increase of 3 °C/min to 180 °C and a further increase of 20 °C/min to 265 °C. This temperature was finally held for 9.25 min. The total running time was 55 min. The temperature in the transfer line was 270 °C. The measurements were performed in single-ion mode (EI^+^, 70 eV). Information on calibration and validation can be found in Appendix A. The five step-calibration was carried out in buffered artificial wine, the validation in a selected wine of the variety Pinot blanc.

### 4.3. Sensory Evaluation

All procedures were conducted in accordance with the relevant legal terms and institutional guidelines, and the participants signed a declaration of agreement.

#### 4.3.1. Panels

An expert and a producer panel, a consumer panel and a descriptive panel were used to assess the typicity of the wines. Table 5 shows the panel composition. The expert and producer panel consisted of Pinot blanc producers or employees of the Höhere Bundeslehranstalt and Bundesamt für Wein- und Obstbau (HBLA and BA für Wein- und Obstbau, Klosterneuburg). The experts were either oenologists or chemists with extensive knowledge of wine production. All experts were familiar with the production of Pinot blanc. Consumers stated that they had consumed wine at least several times a month and Austrian Pinot blanc several times a year.

Regarding the trained descriptive panel, 38 persons attended the training courses. Twelve persons passed the training positively and could participate in the sensory studies. Two odours (citrus and pear dissolved in wine) were used for the discrimination tests. In this case, the test of the triangle with forced choice was used presenting 12 trials, six for each stimulus. A minimum of eight correct answers (α < 0.05) was required to qualify for the study. The trained descriptive panel also consisted of twelve selected persons, including four persons who were members of the expert panel for the evaluation of the wines in 2015 and/or 2016. Table 5 shows the composition of the panels and the classification of the wines evaluated by each panel.

#### 4.3.2. Description of Typical Wines of the “Weißburgunder” (Pinot Blanc) Variety by Different Panels: Sensory Space

A total of 85 Austrian Pinot blanc producers, experts and 198 Pinot blanc consumers participated in this description. All panel members, mentioned in the previous chapter, also participated in this description. The study participants had to fill out a questionnaire on the typicity of Austrian Pinot blanc. The test persons had to answer the question: what are the most typical attributes of Austrian Pinot blanc wine? Typicity was defined as follows: “the important and characteristic features of the aroma of Austrian wines of the Pinot blanc variety for you personally (orthonasal and retronasal perception of the aroma).” From the aroma wheel of the Austrian Wine Marketing Association for Austrian white wine, 41 different aroma terms were available for selection (https://www.austrianwine.com) (Appendix A). The respondents had to choose between three and six aromas. The terms were randomised among subjects within and between panels in the questionnaire.

#### 4.3.3. Panel Training

The trained descriptive panel as well as the expert or producer and consumer panels became familiar with the “sensory space” of wines of the Pinot blanc variety. The word clouds from the description of typical Pinot blanc wines were presented to them (Appendix A). This step served to ensure the existence of a common understanding within the panel regarding the characteristics of a typical Austrian Pinot blanc wine [20]. Before the evaluation of the wines, there was a short technical training for the different panels, and the first two wines served as level wines. The average values of the level wines were read out before being further randomised and tasted.

In contrast to the expert, producer and consumer panel, the trained descriptive panel received further training in a total of nine units of 1.5 hours each. At first, different wines of the variety Pinot blanc were discussed. Both the choice of standards based on the word clouds and the composition of these standards were made by the panel members themselves. The standards were served at room temperature in standard wine glasses wrapped in non-transparent aluminium foil. The standards were improved until they were blindly recognised correctly multiple times by each panel member. The composition of the final standards can be found in Appendix A. In addition to the training sessions and the level wines (described above), the trained panels were blindly served these two level wines three times during the evaluations. There was no significant difference between the results. Apart from that, no further investigations were conducted on reliability and performance.

The banana juice (30% banana pulp), pear juice (50% pear pulp), peach juice (50% peach pulp), apricot juice (50% apricot pulp) and pineapple juice (Happy Day, 100% pineapple) for the training of the trained descriptive panel came from Rauch (Rankweil, Austria). The apple juice (organic apple juice 100%) was purchased from the Höhere Bundeslehranstalt und Bundesamt für Wein- und Obstbau, and the lemons (brand Ja Natürlich) from the local Billa Markt (Klosterneuburg, Austria).

#### 4.3.4. Assessment of Typicity

The tasters evaluated the typicity on an unstructured 10-centimeter scale (from left “atypical” to right “very typical”). They followed the protocol of [15].

#### 4.3.5. Descriptive Analysis of Typical and Atypical Pinot Blanc Wines

From the 50 wines of the 2017 vintage, eight typical Pinot blanc wines (typicity > 6) and eight atypical Pinot blanc wines (typicity < 4) were selected from the average values of consumers and producers or the expert commission. These wines were then quantitatively-descriptively characterized by the trained description panel using an unstructured scale [67,68]. Accordingly, the descriptors developed during the tasting training were used. All wines were analysed in duplicate.

### 4.4. Statistics

The statistical evaluation was done with SPSS 22.0 (IBM, Armonk, NY, USA). The results of the aroma analysis and the typicity assessment were subjected to a univariate analysis of variance with a test for the effects of wine origin, vintage and interaction origin x vintage on three levels of significance (* <0.1; ** <0.01; *** <0.001). The adjustment of the confidence interval was performed by means of an LSD test. The data were previously tested for normal distribution using the Shapiro-Wilk test at the significance level α ≥ 0.05.

Furthermore, a heat map with a dendrogram for the average values of the aromatic concentration of the generic wine regions per vintage in order to estimate whether the vintage or the origin has a wider influence on the aromatic composition. Average linkage was chosen as the cluster method. Euclidean was used as a distance measurement method. Clustering was carried out both in the rows and in the columns. The columns were used as “scale type” [69].

Word clouds were used as a visualisation tool for the raw data of the description of typical Pinot blanc. The data were adjusted to take into account the different group sizes (consumers, producers and experts). Significant differences in the essential attributes (response of at least 10% of respondents in a group) between the groups surveyed were calculated using Cochran’s Q-test.

ANOVA was used to test for significant differences in volatile compounds between wines with low typicity (<4 points) and typical wines (>6 points) of the respective category (vintage or panel evaluation). The typicity categories were analogous to [7], and the data were checked for normal distribution and variance homogeneity. Where the requirements were not met, a Mann-Whitney U test was calculated at the significance level of 0.05. The same approach was used for the different descriptors of the quantitative-descriptive study. By means of non-parametric correlation according to Spearman-Rho, the correlation between the judgements of the three panels from the 2017 vintage was checked.

## 5. Conclusions

Although the complexity of the aroma and its perception makes it difficult to show the connections between typicity and aroma profile, the present study has provided some insights for Austrian Pinot blanc wines. Higher concentrations of ethyl hexanoate, ethyl butanoate, ethyl octanoate, ethyl decanoate, methyl hexanoate, hexyl acetate and isoamyl acetate had a positive effect on the typicity of the wines. A sensory description of typical Pinot blanc wines also showed that the attribute “yellow pome fruit” aroma was significantly more pronounced in typical Pinot blanc wines than in atypical wines. The sensory evaluation of atypical wines showed a stronger expression of the attribute “citrus aroma”. This could be due to higher concentrations of free monoterpenes such as linalool, trans-linalooloxide, neroloxide, nerol and alpha-terpeniol, as higher concentrations of these compounds have led to poorer ratings of the typicity of the wines. Through targeted viticultural and cellar management measures, Austrian winegrowers could use this knowledge to produce more typical Pinot blanc wines, which differ from other origins such as Germany or South Tyrol by a clear concept of typicity.

## Figures and Tables

**Figure 1 molecules-25-05705-f001:**
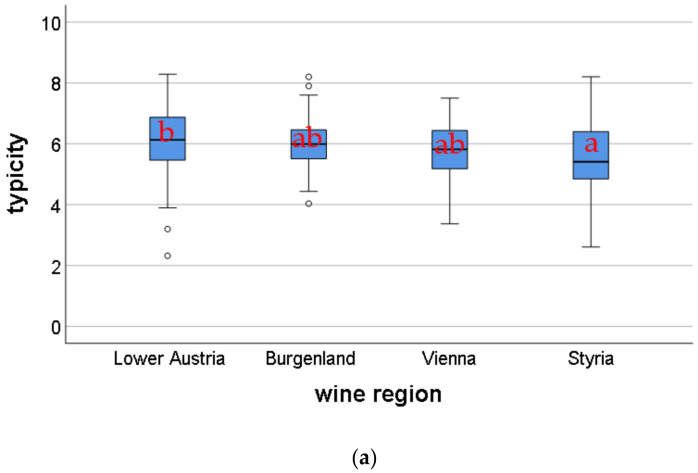
(**a**) Boxplot of the evaluated typicity of Pinot blanc wines from different regions: statistical differences by pairwise comparison of the Mann-Whitney U tests (α ≤ 0.05) after Bonferroni correction; different letters indicate significant differences. (**b**) Boxplot of the assessed typicity of Pinot blanc wines from different vintages; statistical differences by pairwise comparison of the Mann-Whitney U tests (α ≤ 0.05) after Bonferroni correction; different letters indicate significant differences. (**c**) Boxplot of the evaluated typicity of Pinot blanc wines evaluated by different panels; statistical differences by pairwise comparison of the Mann-Whitney U tests (α ≤ 0.05) after Bonferroni correction; different letters indicate significant differences.

**Figure 2 molecules-25-05705-f002:**
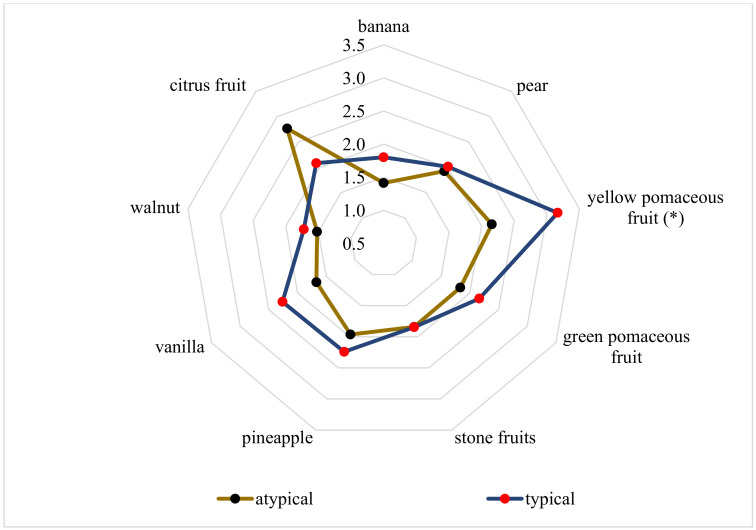
Tasting results of the average value of eight typical and atypical Pinot blanc samples: * indicates a significant difference based on a Mann-Whitney U test (α ≤ 0.05).

**Table 1 molecules-25-05705-t001:** Concentrations of the volatile substance groups.

Volatile Compounds	Mean Values Vintages			Mean Values Origin (Generic Wine Regions)			
	2015	2016	2017	Lower Austria	Burgenland	Styria	Vienna
Free monoterpenes (µg/L) *N* = 8	67.3 ^A^	61.5 ^A^	65.3 ^A^	59.4 ^a,b^	56.2 ^a^	85.2 ^b^	73.0 ^a,b^
C6 compounds (mg/L) *N* = 2	0.85 ^A^	1.74 ^B^	1.21 ^C^	1.38 ^a^	1.20 ^a^	1.33 ^a^	1.15 ^a^
Higher alcohols (mg/L) *N* = 4	351.60 ^A^	216.85 ^A^	223.76 ^A^	222.10 ^a^	213.84 ^a^	210.33 ^a^	239.09 ^a^
Volatile fatty acids (mg/L) *N* = 6	11.75 ^A^	11.71 ^A^	12.32 ^A^	12.04 ^a^	11.84 ^a^	11.57 ^a^	12.52 a
Ethyl esters (µg/L) *^1^ *N* = 11	5504.7 ^B^	4800.3 ^A^	5030.1 ^A,B^	5332.9 ^a^	4781.7 ^a^	4864.2 ^a^	4752.9 ^a^
Methyl esters (µg/L) *N* = 4	11.7 ^A^	14.0 ^B^	15.6 ^B^	14.4 ^a^	13.2 ^a^	12.3 ^a^	14.9 ^a^
Isoamyl esters (µg/L) *n* = 4	1660.4 ^A^	1586.0 ^A^	1856.5 ^A^	1952.1 ^a^	1454.5 ^a^	1415.7 ^a^	1469.6 ^a^
Aromatic esters (µg/L) *N* = 2	2.5 ^A^	2.5 ^A^	2.1 ^A^	2.2 ^a^	2.1 ^a^	3.7 ^b^	1.7 ^a^
Higher alcohol acetates (µg/L) *N* = 3	351.6 ^A^	391.1 ^A^	320.6 ^A^	383.6 ^a^	293.0 ^a^	297.9 ^a^	377.5 ^a^
Mixed and other esters *^2^ (µg/L) *N* = 7	33.2 ^A^	39.2 ^B^	30.8 ^A^	34.0 ^a^	35.7 ^a^	33.4 ^a^	35.6 ^a^
C13-nonorisoprenoids (µg/L) *N* = 2	2.1 ^A^	1.7 ^A^	1.7 ^A^	1.8 ^a^	2.1 ^a^	1.4 ^a^	1.9 ^a^
Carbonyl compounds (µg/L) *N* = 6	118.0 ^A^	101.9 ^A^	162.3 ^A^	118.2 ^a^	162.2 ^a^	125.5 ^a^	132.4 ^a^
Lactones (µg/L) *N* = 3	153.3 ^A^	142.8 ^A^	194.8 ^A^	154.0 ^a^	200.0 ^a^	160.3 ^a^	169.9 ^a^
Volatile phenols (µg/L) *N* = 12	193.8 ^A^	194.0 ^A^	171.5 ^A^	186.1 ^a^	197.2 ^a^	139.1 ^a^	214.6 ^a^

*^1^ excluding ethyl lactates and ethyl acetates; *^2^ excluding diethyl succinate. Different uppercase letters indicate significant differences in the mean values of the different vintages. Different lowercase letters indicate significant differences in the mean values of origin.

**Table 2 molecules-25-05705-t002:** Typical attributes for Austrian Pinot blanc: relative frequency of mention by consumers and producers or experts.

Attribute Group	Attributes	Relative Frequency of Naming by Consumers (*n* = 198)	Relative Frequency of Naming by Producers and Experts (*n* = 85)
Stone fruit	Apricot	13 ^b^	4 ^a^
	Peach	10 ^a^	9 ^a^
Pome fruit	Green apple	19 ^b^	10 ^a^
	Yellow apple	31 ^b^	13 ^a^
	Pear	22 ^a^	25 ^a^
Nut	Walnut	7 ^a^	19 ^b^
	Almond	7 ^a^	15 ^b^
Citrus Exotic fruits	Lemon	15 ^b^	7 ^a^
	Pineapple	15 ^a^	14 ^a^
	Banana	6 ^a^	9 ^b^
Flowery	Elderflower	12 ^b^	1 ^a^
	Meadow flowers	0 ^a^	15 ^b^
Wood	Vanilla	23 ^b^	11 ^a^
	Oak	2 ^a^	12 ^b^

Significant differences (indicated by different letters) in the essential attributes (response of at least 10% of respondents in a group) between the groups surveyed were calculated using Cochran’s Q-test (α ≤ 0.05).

**Table 3 molecules-25-05705-t003:** Volatile compounds found significantly (α ≤ 0.05) in typical Pinot blanc wines compared to atypical wines in higher quantities, ranked by frequency of significant difference.

Wines	A	B	NO	ST	W	2015	2016	2017		
Panel	E	E	E	E	E	E	E	E	P *^1^	K *^1^
ethyl hexanoate	x		x		x	x	x	x	x	x
ethyl butanoate	x		x		x			x	x	x
ethyl octanoate	x				x		x	x	x	x
ethyl decanoate	x				x			x		x
methyl hexanoate	x		x					x	x	
hexyl acetate	x	x	x						x	
methyl decanoate					x			x	x	
isoamyl acetate	x				x				x	
isoamyl octanoate						x	x			x
methyl vanillate			x			x	x			
methyl octanoate								x	x	
propyl octanoate								x		x
delta-decalactone							x			x
1-butanol					x					
(*Z*)-3-hexen-1ol			x							
1-hexanol						x				
ethyl acetate									x	
ethyl dodecanoate										x
diacetyl						x				
guaiacol									x	
acetovanillone			x							

*^1^ Medium typical and atypical combined into one category because there are too few atypical wines; A = all wines, B = Burgenland, NO = Lower Austria; ST = Styria; W = Vienna; 2015 = wines from vintage 2015; 2016 = wines from vintage 2016; 2017 = wines from vintage 2017; E = expert panel, P = trained descriptive panel; K = consumer panel.

**Table 4 molecules-25-05705-t004:** Volatile compounds found in atypical wines in higher quantities compared to typical Pinot blanc wines (α ≤ 0.05).

Wines	All	B	NO	ST	W	2015	2016	2017		
Panel	E	E	E	E	E	E	E	E	P *^1^	K *^1^
linalool	x					x	x	x		
isovaleric acid				x				x	x	x
hexanoic acid				x				x	x	x
(*E*)-linalool oxide	x							x	x	
nerol oxide	x							x	x	
alpha-terpineol	x							x	x	
nerol	x							x	x	
isobutanol	x							x	x	
isoamyl alcohol				x				x	x	
isobutyl octanoate	x							x		
butyl butanoate						x	x		x	
(*E*)-whiskey lactone	x			x					x	
hotrienol	x							x		
ethyl isovalerate								x		
ethyl phenylacetate								x	x	
2-methoxy-4-propylphenol								x	x	
(*E*)-Isoeugenol	x				x					
1-propanol				x						
butyl isobutanoate								x		
2-methoxy-4-propylphenol	x									
eugenol					x					
4-methylguaiacol				x						

*^1^ Medium typical and atypical combined into one category because there are too few atypical wines; A = all wines, B = Burgenland, NO = Lower Austria; ST = Styria; W = Vienna; 2015 = wines from vintage 2015; 2016 = wines from vintage 2016; 2017 = wines from vintage 2017; E = expert panel, P = trained descriptive panel; K = consumer panel.

**Table 5 molecules-25-05705-t005:** Panel overview.

Wines	Panel of Experts and Producers	Consumer Panel	Trained Descriptive Panel
Wines vintage 2015 (*n* = 36)	yes (3 women, 7 men, age: 21–91)	no	no
Wines vintage 2016 (*n* = 45)	yes (3 women, 5 men, age: 25–54)	no	no
Wines vintage 2017 (*n* = 50)	yes (2 women, 5 men, age: 24–56)	Yes (50 women, 55 men, age: 19–72)	yes (6 women, 2 men: age: 22–46)
Description of typical Pinot blanc wines	yes (31 women, 54 men; age: 20–91)	Yes (89 women, 109 men, age: 18–78)	no

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
