# Peer review of "Aromatypicity of Austrian Pinot Blanc Wines"

_molecules, 2020, doi:10.3390/molecules25235705_

Round 1

Reviewer 1 Report

General comments

The manuscript investigates the relationship between the aroma composition and the perception of typicity of a high number of Austrian Pinot blanc wines.  The topic is interesting and worthy of study. The manuscript has appropriate introduction, results and discussion. However, the main concern that I have with the work is that the study involved the use of human subjects but no statements are included in the manuscript regarding the ethical issues. Since there are some relevant concerns that I have with the work (which are described below), I do not recommend the publication of the manuscript in the present form.

Specific comments

L89-91. Please delete. Those are results, not appropriate for the Introduction section.

L176. I would suggest to refer to Fig. 1A instead to Appendix A

L188. Please specify in the table footnote the statistical test applied to associate the different letters to the mean values.

L217. How many typical wines and how many atypical wines were considered?

L226-228. The terms “very(typical)” and “(medium) atypical” are note defined in the text, where just the terms “typical” and “atypical” were defined based on the typicity value (>6 and >4). Please revise.

L236. Please clarify this statement. Do you mean that higher concentrations of linalool were found to be rather negative because the typicity was considered low?

L253 and 258. Please delete “Descritpive”.

L406. Please change “Sensor study” to “Sensory evaluation”.

L407. Please change “Panel” with “Panels”

L415. Did you check the trained panel performance to verify the reliability of the collected data? How? Moreover, descriptive data are usually analyzed with a three-way anova (random factor: subjects; fixed factors: wines, replicates) but I could not see it in the manuscript. Please provide an explanation fir that or add it.

L408-418. Please, provide more details about the subjects composing the four panels. Please add the % of males and females of the participants, their mean age and standard deviation.

The study involved the use of human subjects but no statements are included in the manuscript regarding the ethical issues. I assume that the protocol of the study was approved by an ethical committee, all procedures were performed in compliance with relevant laws and institutional guidelines and that the participants signed an informed consent. Please clarify and include appropriate ethical statements and mention the approval by a specific ethical committee.

L429. Please specify the selection criterion given to the subjects (most typical attributes for Austrian Pinot blanc?).

L430. Were the terms randomized among subjects within and between panels?

L439. Change “the expert, producer and consumer panel” with “the expert, producer and consumer panels”

L442. Please provide information about how the standards were presented to the panel members (e.g in black glasses; at which temperature?)

L451-454. It is not clear to me which were the verbal anchors of the extremes of the scale: “atypical” and “very typical” or “good example“ and “bad example”?

L456. Indicate how many typical and atypical wines were selected. Out of 131 analysed, 8+8 (16) wines were selected. Were they representative oof the totality of the available wines? Please provide more details on the criteria used to select them. I strongly recommend to report also how many wines resulted belonging to each of the three typicity classes: <4, >6, between 4 and 6.

Table S1: Change “oring” to “origin”

Figures 1a, b,c. Please display the values of the Y-axis without decimal places.

Figure S3 and S4. Please indicate the significance (stars) of the attributes. Alternatively, I would suggest to convert the radar plots to histograms showing letters for significant differences.

Tables. Please use commas instead of dots as decimal separator.

Author Response

Reply to Review Report 1

First of all we would like to say thank you. Your constructive comments have significantly improved our manuscript.

Our comments to your suggestions.

General comments

The manuscript investigates the relationship between the aroma composition and the perception of typicity of a high number of Austrian Pinot blanc wines.  The topic is interesting and worthy of study. The manuscript has appropriate introduction, results and discussion. However, the main concern that I have with the work is that the study involved the use of human subjects but no statements are included in the manuscript regarding the ethical issues. Since there are some relevant concerns that I have with the work (which are described below), I do not recommend the publication of the manuscript in the present form.

Thank you very much for your comments. We have included a statement to this respect in the manuscript. All processes are of course in accordance with the law, and institutional rules are also observed. The participants have signed a declaration of consent. Currently there is no Ethic Committee at our small institute, but all projects are controlled and approved by the Ministry. Of course, ethical aspects are controlled during this review.

Specific comments

L89-91. Please delete. Those are results, not appropriate for the Introduction section.

We deleted this part of the Introduction.

L176. I would suggest to refer to Fig. 1A instead to Appendix A

Ok I changed to Fig. 1A

L188. Please specify in the table footnote the statistical test applied to associate the different letters to the mean values.

I did so

L217. How many typical wines and how many atypical wines were considered?

I added a table in the supplementary data to show how many typical/untypical and medium typical wines have been found by the different panels.

L226-228. The terms “very(typical)” and “(medium) atypical” are note defined in the text, where just the terms “typical” and “atypical” were defined based on the typicity value (>6 and >4). Please revise.

Your are right, this is very complicated I delated the term very typical and indicated medium typical

L236. Please clarify this statement. Do you mean that higher concentrations of linalool were found to be rather negative because the typicity was considered low?

We have added a sentence that clarifies the statement. Higher concentrations of linalool resulted in lower typicity ratings.

L253 and 258. Please delete “Descritpive”.

I deleted the word

L406. Please change “Sensor study” to “Sensory evaluation”.

I did so.

L407. Please change “Panel” with “Panels”

I changed to panels.

L415. Did you check the trained panel performance to verify the reliability of the collected data? How? Moreover, descriptive data are usually analyzed with a three-way anova (random factor: subjects; fixed factors: wines, replicates) but I could not see it in the manuscript. Please provide an explanation fir that or add it.

I added information regarding the panel performance. I was not able to calculate a three-way anova, because the data was not normal distributed. This is a statistical assumption. I have included this information in the manuscript so that no one is surprised. Thanks for the very important notice.

L408-418. Please, provide more details about the subjects composing the four panels. Please add the % of males and females of the participants, their mean age and standard deviation.

I added some information to the table.

The study involved the use of human subjects but no statements are included in the manuscript regarding the ethical issues. I assume that the protocol of the study was approved by an ethical committee, all procedures were performed in compliance with relevant laws and institutional guidelines and that the participants signed an informed consent. Please clarify and include appropriate ethical statements and mention the approval by a specific ethical committee.

 See above

L429. Please specify the selection criterion given to the subjects (most typical attributes for Austrian Pinot blanc?).

I did so

L430. Were the terms randomized among subjects within and between panels?

This is a good point, we added the information.

L439. Change “the expert, producer and consumer panel” with “the expert, producer and consumer panels”

I did so.

L442. Please provide information about how the standards were presented to the panel members (e.g in black glasses; at which temperature?)

Yes, I added some information.

L451-454. It is not clear to me which were the verbal anchors of the extremes of the scale: “atypical” and “very typical” or “good example“ and “bad example”?

You are right. The spelling was very confusing. I deleted two sentences. Now it should be clearer.

L456. Indicate how many typical and atypical wines were selected. Out of 131 analysed, 8+8 (16) wines were selected. Were they representative oof the totality of the available wines? Please provide more details on the criteria used to select them. I strongly recommend to report also how many wines resulted belonging to each of the three typicity classes: <4, >6, between 4 and 6.

We understood your objection and tried to explain the selection more transparently. This was necessary, thank you very much for the remark. Only wines of vintage 2017 were available for this attempt, since we had installed a trained panel only for this vintage. So in total 16 samples out of 50 were sensory examined more closely. For the selection we took the average value of the consumer and producer panel. This resulted in exactly 8 typical and 8 untypical wines. The validity of these results is due to the good agreement with the trained panel. The question whether 8 typical wines are representative of all typical Pinot Blanc wines or whether 8 atypical wines are representative of all atypical wines cannot be answered easily. I dare a direct comparison with other published studies. Comparing the number of samples, it is noticeable that we analyzed 131 samples, which is far beyond the normal sample size. Other studies examine 10-20 samples and do all sensory studies also with exactly these 10-20 samples. We think that we have explained the selection in a transparent way, so that the readers of the publication can get an idea for themselves.

Table S1: Change “oring” to “origin”

I did so

Figures 1a, b,c. Please display the values of the Y-axis without decimal places.

I did so

Figure S3 and S4. Please indicate the significance (stars) of the attributes. Alternatively, I would suggest to convert the radar plots to histograms showing letters for significant differences.

I did so.

Tables. Please use commas instead of dots as decimal separator.

Since another reviewer asked for the opposite, we followed the English spelling and used dots instead of commas.

Thanks again for your comments

KR

Dr. Christian Philipp

Reviewer 2 Report

The manuscript entitled “Aromatypicity of Austrian Pinot blanc wines” characterize the typicity of Austrian Pinot blanc wines with regard to their aroma profile. For this purpose wines of different vintages and origins were analysed. This manuscript should be accepted in Molecules after minor revisions.

All abbreviations should be described when reported for the first time.

The authors should adopt the same units (sometimes used µl other µL; ml other mL) and language (sometimes minutes other min).

In english the decimal is point instead comma (Table S1, S4)

Line 127: 1 hexanol/(T)-3-hexen-1ol should be 1-hexanol/(Z)-3-hexen-1ol

Line 153: please indicate the limit of quantification

Line 369: “3.4 - dimethylanisole” should be “3,4 - dimethylanisole

Line 386: 1,1,6 trimethyl-1,2-dihydronaphtaline should be 1,1,6 trimethyl-1,2-dihydronaphtalene
